# Polyethylene Oxide Assisted Fish Collagen-Poly-ε-Caprolactone Nanofiber Membranes by Electrospinning

**DOI:** 10.3390/nano12060900

**Published:** 2022-03-09

**Authors:** Xiaoli He, Lei Wang, Kangning Lv, Wenjun Li, Song Qin, Zhihong Tang

**Affiliations:** 1Coastal Zone Biology and Biological Resources Protection Laboratory, Yantai Institute of Coastal Zone Research, Chinese Academy of Sciences, Yantai 264003, China; hexiaoli97@163.com (X.H.); bio_wangl@163.com (L.W.); lvkangning2018@163.com (K.L.); wjli@yic.ac.cn (W.L.); 2College of Life Sciences, Yantai University, Yantai 264005, China

**Keywords:** fish collagen, electrospinning, nanofiber, spinning aid

## Abstract

Fish collagen has higher biocompatibility and lower immunogenicity than terrestrial collagen, and is currently one of the important raw materials for preparing biological materials. In this study, PEO was used as a spinning aid to prepare fish skin collagen-PCL nanofiber membranes by electrospinning, and the process was optimized to get smooth nanofibers. The morphological and mechanical properties of collagen-PCL nanofiber membranes were assessed by scanning electron microscopy (SEM). The changes in chemical composition due to the incorporation of collagen into PCL and PEO were determined by Fourier Transform infrared spectroscopy (FTIR). The biocompatibility of the collagen-PCL nanofiber membranes was evaluated in vitro in cultures of mouse fibroblasts and in vivo by subcutaneous implantation studies in rats. It was found that the diameter of the spun fibers became fine and smooth when the ratio of the collagen/PCL increased. The finally obtained nanofiber had good mechanical strength, porosity, and hydrophilicity, and could promote cell adhesion and proliferation. The FC-PCL nanofiber membrane prepared by this route opens a new way to prepare fish collagen biomaterials with electrospinning.

## 1. Introduction

Collagen is the most abundant protein in the human body and has become one of the most widely used materials in tissue engineering due to its superior biological properties [1]. Compared with land-based collagen, collagen derived from aquatic animals has the advantages of a wide range of sources, low production cost, low immunogenicity, and is free from the influence of mad cow disease, religion, and other factors [2], so it has been applied more and more widely in the field of materials. Tilapia (*Oreochromis mossambicus*) is a common freshwater fish in China. The processing of Tilapia can produce a lot of fish skin, which will pollute the environment if it is not handled properly. Therefore, the extraction of collagen from Tilapia skin (FC) and further development and utilization can not only increase the added value of products, but also avoid resource waste and reduce environmental pollution [2]. It has also been reported that the Tilapia skin collagen film-forming property is significantly better than that of freshwater fish such as carp (*Hypophthalmichthys molitrix*) and grass carp (*Ctenopharyngodon idella*) [3].

In practical application, pure collagen material has problems such as low thermal stability [4], poor mechanical properties, and fast degradation rate. Therefore, the question of how to effectively prepare collagen material with excellent properties has become a research hotspot. Studies have shown that when the fiber diameter of the material is changed from micron to submicron or nanometer scale, these nanofibers could simulate the size, structure, and function of natural extracellular matrix components, and have better biocompatibility and mechanical strength [5]. At present, the methods to fabricate collagen-based nanomaterials include drafting, template polymerization, phase separation, self-assembly, and electrospinning [6], etc. Compared with other methods, electrospinning has the advantages of simple devices, low costs, and a controllable process. The diameter of collagen fibers prepared by electrospinning technology can reach micron or even nanometer level, and the membrane material formed has the characteristics of high porosity, good homogeneity, and a large specific surface area, which is the ideal material for tissue engineering scaffolds [7].

In order to improve the performance of collagen materials, in the electrospinning process, collagen is usually combined with other polymer materials, such as chitosan, polyacrylic acid (PAA), polyvinyl alcohol (PVA), poly-ε-caprolactone (PCL), polyvinylpyrrolidone (PVP), and polyethylene oxide (PEO) to prepare biological materials with good mechanical properties and stability [8]. PCL is an FDA-approved polymer for clinical use. However, it is difficult to degrade in vivo. By using electrospinning technology, fish collagen protein and PCL can be blended. The collagen-PCL composite membrane prepared has both the excellent biological properties of collagen and the good mechanical energy of PCL [9]. Park, S. et al. [10] prepared double-layer vascular grafts using PCL/collagen and PCL/silica nanoparticles with good biocompatibility and mechanical strength. Metwally, S. et al. [11] found that PCL fibers added with collagen were more conducive to cell adhesion and proliferation.

PEO is one of synthetic polymer used in the basic research of the electrospinning process [12]. It reduces the surface tension of the spinning liquid, facilitates the splitting of the fibers, and makes the diameter distribution of the fibers more uniform. Tenchurin, T.K. et al. [13] added PEO to collagen dispersion to improve spinnability. However, no electrospinning PEO fibers have been used as the substrate and scaffold for cell cultures up to now, mainly because pure PEO fibers without cross-linking can dissolve quickly in a water environment and protein absorption is very weak, making it difficult for cells to adhere to the PEO surface [14]. Therefore, in the electrospinning process, the PEO fiber can be removed by the template sacrificial method. This method increases the porosity of the membrane material and promotes cell adhesion and proliferation [15]. However, there has not been any report on using PEO to assist the spinning of fish collagen-PCL.

The objective of this project is to optimize the preparation of FC/PCL nanofiber membranes, using PEO as the sacrificial fiber to determine the effect of its removal on the characterization of the prepared nanofiber materials. All animal studies were conducted in general accordance with the guidelines of the Institutional Animal Care and Use Committee (IACUC).

## 2. Materials and Methods

### 2.1. Materials and Equipment

Tilapia skin collagen (FC) was obtained from Yantai Langdi Biotechnology Co. Ltd. (Yantai, China). Polycaprolactone (PCL, MW = 80,000) was purchased from Shanghai yuanye Bio-Technology Co. Ltd. (Shanghai, China). Glacial acetic acid (HAc, AR), polyvinyl oxide (PEO, MV =100, 000) was bought from Shanghai Macklin Biochemical Co. Ltd. (Shanghai, China). Hexafluoro isopropanol (HFIP, 99.5%) was purchased from Shanghai Aladdin Biochemical Technology Co. Ltd. (Shanghai, China). The electrospinning instrument is NANON-01 (MECC, Fukuoka, Japan), Contact angle tester (SDC-500).

### 2.2. Preparation of FC-PCL Nanofiber Membranes

#### 2.2.1. PEO Spinning

Based on the method of Chakrapani, V.Y. et al. [16], the addition of PEO was optimized. HFIP and HAc were mixed at a volume ratio of 1:1, and a mixture of FC and PCL (FC:PCL = 8:2) was added to prepare 10 wt% electrospinning solution. In addition, an appropriate amount of PEO (0.2%, 0.4%, 0.6%, 0.8%, 0.1%, 0.12%) was added as a spinning aid. The solution was stirred until there were no obvious solid particles.

The electrospinning solution was then put into a syringe, connected to a 21-gauge blunt needle, and installed on the syringe pump. The process parameters (voltage, flow rate, receiving distance, chamber humidity, and needle-collector distance) were varied to optimize the stability of the electrostatic jet during electrospinning. Preferred conditions were found to be a spinning voltage of 18 kV, a receiving distance of 11 cm, a solution advancing speed of 0.8 mL/h, and spinning chamber humidity of 50%. After adding different qualities of PEO to aid spinning, the morphology of the receiving fiber material was observed.

The prepared FC-PCL nanofiber membranes were placed in a constant temperature drying oven at 37 °C for 48 h to remove residual solvents.

#### 2.2.2. FC/PCL Ratio

FC and PCL with ratios of 0:10, 1:9, 2:8, 3:7, 4:6, 5:5, 6:4, 7:3, 8:2, 9:1, 10:0, respectively were formulated into 10 wt% electrospinning solution, and a certain amount of PEO was added as a spinning aid. The preparation steps were the same as above. The fiber state was observed by SEM, and the swelling rate and hydrolytic stability of the material were tested.

#### 2.2.3. Removal of Sacrificial Fibers of PEO

The samples were rinsed with PBS 3–5 times, and the PEO sacrificial fibers were removed from the electrospinning samples by shaking in a constant temperature shaking incubator overnight, and freeze-dried. The mass loss was calculated based on the mass before and after PEO removal.
$$PEO\;theoretical\;mass=mass\;of\frac{sample*PEO\;content}{FC\;content+PCL\;content+PEO\;content}$$
Sample loss mass=mass of sample before cleaning−mass of sample after cleaning

### 2.3. Physical and Chemical Properties Testing

#### 2.3.1. SEM

The FC-PCL nanofiber films prepared by different mass ratios of FC and PCL were fixed on the sample platform by conducting adhesive. After gold was sprayed on the surface, the microstructure of the sample fibers was observed under an S-4800 cold-field emission scanning electron microscope. SDM software was used to measure the diameters of 50 different fibers, and their arithmetic mean values were obtained.

#### 2.3.2. Swelling Rate

The lyophilized nanofiber membrane was weighed (*M*), immersed in PBS (pH = 7.4), and stored at 37 °C. The membrane was taken out per 2 h, and the excess water on the surface was gently wiped with a filter paper and weighed (*M*_1_, *M*_2_… *M*_max_) until no significant change was observed for three consecutive times. The swelling ratio was calculated as Equation (1):(1)$$Qs=\frac{M_{max}-M}{M}\times 100\%$$
where *Qs* is the swelling ratio of the membrane, *M*_max_ and *M* are the maximum wet weight and initial dry weight of the collagen nanofiber membrane, respectively.

#### 2.3.3. Hydrolysis Stability

The collagen nanofiber membrane was lyophilized and weighed (*M*), then immersed in PBS (pH 7.4), stored at 37 °C for 96 h, then the sample was taken out, freeze-dried, and weighed (*Mz*). The solubility was calculated using Equation (2):(2)$$Es=\frac{M-Mz}{M}\times 100\%$$
where *Es* is the hydrolysis stability of the membrane, *Mz* is the dry weight of the membrane after 96 h immersion in PBS, and *M* is the dry weight of the initial membrane [17].

#### 2.3.4. Contact Angle

The sample was cut into 1 cm × 1 cm slices and fixed on the slides. The contact angles of different spinning films were detected by SDC-500 contact angle measuring instrument and water drop shape analysis system.

#### 2.3.5. FTIR

The chemical structures of FC-PCL nanofiber membranes before and after PEO removal were analyzed by Nicolet 6700 Fourier transform infrared spectroscopy (FTIR, Thermo Fisher, Waltham, MA, USA), and the scanning range was 400–4000 cm^−1^.

#### 2.3.6. Tensile Breaking Strength Test

At room temperature, the FC-PCL nanofiber membrane was made into a regular sample of 5 mm × 50 mm. The sample was placed in a fixture and tested with a universal tester. The maximum load was 200 cN, the sensitivity was 0.01 cN, and the test speed was 5 mm/min. The thickness of the FC-PCL nanofiber membrane was measured with a spiral micrometer.

#### 2.3.7. Porosity

According to the method of Aghmiuni, A.I. et al. [7], the porosity of the scaffold was measured by liquid displacement method. Using absolute ethanol as the replacement fluid, the holder, empty pycnometer, and ethanol were kept at 25 °C for 1 h. The pycnometer with absolute ethanol was weighted (*W*_1_). In addition, the lyophilized stent (*Ws*) of known weight was slowly immersed in the bottle and refilled with ethanol, and weighed after 5 min (*W*_2_). The saturated holder containing ethanol was removed from the bottle, and the weight of the pycnometer(*W*_3_) was recorded. Finally, Equations (3) and (4) were used to obtain the volume of the scaffold (*Vs*) and the total volume of pores (*Vp*), and Equation (5) was used to determine the porosity of the scaffold.
(3)$$V_{s}=\frac{W_{1}-W_{2}+W_{3}}{\rho }$$
(4)$$V_{p}=\frac{W_{2}-W_{3}-W_{s}}{\rho }$$
(5)$$\zeta =\frac{V_{p}}{V_{p}+V_{s}}=\frac{W_{2}-W_{3}-W_{s}}{W_{1}-W_{3}}$$
ρ and ζ are the density of anhydrous ethanol and the porosity of the scaffold respectively.

### 2.4. Cell Culture

#### 2.4.1. Cell Adhesion

L929 cells were cultured in DMEM high glucose complete medium containing 10% fetal bovine serum and 1% penicillin and streptomycin antibiotics. The cells were placed in a 37 °C, 5% CO_2_ cell incubator, digestied, and counted when they grew to logarithmic growth phas. 0.5 × 104 cells/well was used as the standard to inoculate sterile samples. After the material and the cells were co-cultured for 24 h, they were fixed with 2.5% glutaraldehyde for 24 h. After freeze-drying, the cell infiltration and growth were observed under a scanning electron microscope.

#### 2.4.2. Dead/Live Cell Text

L929 was inoculated in 48-well plates at a density of 5 × 10^4^ cells/well. After 24 h of culture, the medium was removed and scaffold extract was added for the experimental group. Conventional DMEM medium was set as the control group. There were 5 replicate wells in each group, stained with AMPI kit on 1, 3, and 5d respectively, and the status of living and dead cells was observed under a fluorescence microscope (Echo, Lake Zurich, IL, USA).

### 2.5. Safety and Degradation Tests In Vivo

Twenty-five 8-week-old male SD rats were implanted with nanofiber membrane with FC PCL ratio of 6:4 (FP 6-4). After anesthesia, they were implanted subcutaneously with 0.5 cm × 0.5 cm film. At 3/7/14/21/28 d after modeling, 5 rats were randomly selected at each time node, the subcutaneous implanted tissue was taken out and fixed in 10% paraformaldehyde solution. Tissue sections were stained with hematoxylin and eosin (H & E staink) and examined using optical microscopy [18].

### 2.6. Statistical Analysis

All the experiments were performed in triplicates, and the data are presented as mean ± standard deviation.

## 3. Results and Discussion

### 3.1. Preparation of FC-PCL Nanofiber Membrane

#### 3.1.1. PEO Spinning

Pore size and bulk density are important factors that affect the cell infiltration and mass transfer of biological materials. However, one of the limitations of the electrospinning process is the dense accumulation of fibers, and the small pore size will inhibit cell infiltration. Therefore, there have been many studies using PEO as a sacrificial fiber to increase the pore size of the material. Shin, J.W. et al. [18] sacrificed PEO fiber to prepare a multilayer polyurethane (PU) scaffold, which had a higher performance than a single PU scaffold. Hodge, J. et al. [19] found that PEO sacrificial fibers could improve the porosity and cell infiltration of polymer materials prepared by electrospinning, such as polyglycolic acid (PGA), polylacti acid-glycolic acid copolymer (PLGA), and polycaprolactone (PCL). Besides, the addition of PEO to the electrospinning solutions improved their electrical conductivity, surface tension, and viscosity, greatly favoring the electrospinning process [20].

In the present study, the effects of different PEO additions on spinning were investigated (Table 1). When the PEO addition was less than or equal to 0.4%, a large amount of liquid drips would appear during the electrospinning process, and a complete film couldn’t be received. When the content of PEO exceeded 1.0%, the solution gradually became unstable. After some fibers appeared, liquid began to drip, and a complete film couldn’t be received under this electrospinning condition. When the amount of PEO was 0.6–0.8%, the complete membrane could be received. The purpose of this experiment is to use PEO as sacrificial fiber to obtain fiber membrane with large porosity. Therefore, the final dosage of PEO spinning aid was 0.8%.

#### 3.1.2. FC/PCL Ratio

The structure of nanofiber membranes with different FC and PCL ratios could be observed by SEM, and the fiber and pore diameters were analyzed through SDM software. It was shown in Figure 1a–k that the fiber state of the blend membrane gradually improved with the decrease of PCL ratio and the increase of FC ratio. When the content of PCL exceeded 80%, the received material was difficult to distinguish the fiber structure, and there were almost no pores in the material (Figure 1a–c). With the addition of collagen, the fibers gradually became clear, and the fibrous structure began to appear (Figure 1c). It can be seen from Figure 1d–g that the fiber structure became increasingly obvious, and the porosity gradually rose as the FC ratio increased. When the content of FC exceeded 70%, the fiber structure basically tended to be stable, relatively smooth, and uniform fibers could be formed, and the fiber diameter reached the nanometer level (Figure 1h–k).

The fiber diameter was analyzed by SDM and Origin software (Figure 2). With the increase of collagen content, the average fiber diameter became smaller, but an exception occurred when m (FC): m (PCL) was 6:4. In this case, the state of the solution might be unstable and there was adhesion between the fibers, resulting in the formation of very fine fibers. Therefore, the fiber diameter changed greatly and the average fiber diameter decreased. Figure 2 showed that the higher the proportion of collagen, the more uniform the fiber diameter distribution. When m (FC): m (PCL) was 10:0, the diameter was mainly distributed between 100–150 nm. When m (FC): m (PCL) was 9:1, the diameter was mainly distributed between 150–250 nm. With the decrease in collagen ratio, the fiber diameter distribution gradually dispersed.

#### 3.1.3. Removal of Sacrificial Fibers of PEO

A good biomaterial should have enough spatial structure to support cell growth. Therefore, it is very important to maintain a stable spatial structure of the material after PEO removal. Figure 3a,b showed that after PEO removal, the spatial structure of the material is still very complete, and only some “fine fibers” were reduced.

When rinsing with PBS to remove PEO, the actual mass loss of the sample should be greater than the theoretical mass of the added PEO if PEO can be completely removed and partially dissolved collagen can be added. This result occurs when the collagen ratio is greater than 20% (Figure 3c). However, when the proportion of collagen was less than 20%, the actual mass loss of the sample was less than that of PEO, which may be due to the dense structure, which made it difficult to remove PEO from the sample. However, in the membrane with high collagen content, it was impossible to accurately judge whether the mass loss was due to the removal of PEO or the dissolution of most collagen. Therefore, the results need to be further analyzed by infrared spectroscopy.

### 3.2. Physical and Chemical Properties Testing

#### 3.2.1. FTIR

The infrared spectra were shown in Figure 4, and the absorption bands of different pure polymers (FC, PCL, PEO) were summarized in Table 2. The absorption peak near 1238 cm^−1^ was the absorption peak of the amide III band, which was caused by the C-O stretching vibration [11], which was also related to the triple helix structure of collagen [21]. The characteristic peaks of type I collagen was more obvious after PEO removal. This result may be due to the unstable “crosslinking” of PEO and collagen before material cleaning. After PEO was removed by cleaning, more collagen functional groups were exposed.

Characteristic bands of PCL of the carbonyl groups associated with the ester bonds were found at 1728 cm^−1^ [11]. When free N-H participates in the formation of hydrogen bonds, its stretching vibration will shift to a low frequency. In the infrared absorption curve of FC-PCL nanofiber membrane, the C=O vibration peaks of PCL and the C=O, N-H [19], and C-H vibration peaks of collagen were red shifted (Figure 4a,c,d), indicating that there were new hydrogen bonds between the amide group of collagen and the carboxyl group of PCL [22].

The CH_2_ tensile vibration absorption peaks of PCL and PEO were located near 2880 cm^−1^, so the relative intensity of the infrared absorption peak of the film at this wavelength increased and widened (Figure 4a,b,d). After PEO removal, the relative intensity of CH_2_ tensile vibration absorption peak decreased. In addition, the relative intensity of the C-O-C tensile vibration absorption peak of the membrane decreased at 1148 cm^−1^ and 958 cm^−1^, and the C-H deformation of methyl also decreased at 845 cm^−1^ and 1340 cm^−1^ (Figure 4b,d,e). According to the quality of PEO removal and the change of infrared absorption peak, it can be preliminarily judged that PEO removal is successful.

#### 3.2.2. Swelling Rate and Hydrolysis Stability

During the experiment, pure collagen nanofiber membrane dissolved rapidly in physiological liquids such as PBS, and didn’t have hydrolytic stability. Since the solubility of pure collagen nanofiber membrane was 100%, if PCL does not bind to collagen, the weight of the blend membrane will be reduced by at least the same amount as FC. However, in the actual process, the weight loss of the blend membrane was much less than that of FC, which indicated that there was a certain “reaction” between PCL and FC, which reduced the solubility of collagen. FTIR results also confirmed this conjecture.

When the collagen content exceeded 80%, the expansion rate of the membrane decreased abnormally and the solubility decreased in a cliff-like manner (Figure 5a), which may be related to the number of hydrogen bonds formed between PCL and FC. With the decrease in PCL content, most FC still existed in the form of free collagen. Because the hydrolysis stability of collagen was poor, when m (FC): m (PCL) = 9:1, a large amount of collagen in the material was dissolved. The hydrophilicity of PCL was much lower than that of collagen, resulting in the greatly reduced expansion rate of the membrane.

Since most soluble collagen and unreacted PEO were removed by cleaning, the hydrolysis stability of the cleaned membrane was higher than 85%. However, the collagen content was still slightly decreased when it exceeded 60%. This may be due to there being fewer hydrogen bonds between collagen and PCL, resulting in the loss of collagen after cleaning. When the collagen content exceeded 60%, the hydrolysis stability of the material fluctuated very little. When m (FC): m (PCL) = 6:4, the expansion rate of the material was the highest (Figure 5b). Biomaterials should have good biological properties, mechanical properties, and spatial structure to support cell growth. In addition, the implanted material should also have certain hydrolysis stability to prevent excessive degradation in vivo. Considering the above factors, we selected m(FC): m(PCL) = 6:4(FP 6-4) nanofiber membrane as the representative material for biological evaluation and characterized it.

Biomaterials will come in contact with body fluids, so the wettability of biomaterials must be characterized for the preparation of biomaterials, which is generally accomplished by evaluating the contact angle formed by the liquid on the surface of the electrospinning substrate [16]. Good biomaterials need high hydrophilicity, that is, contact angle <90°, which is conducive to cell adhesion. Tu, H.B. et al. [23] used collagen coating to increase the hydrophilicity of the material, and the contact angle decreased from 104.52 ± 4.09° to 47.93 ± 2.09° after coating the collagen solution on the electrospinning PLLA scaffold material. Although PEO has hydrophilic properties, it has poor hygroscopic properties and is not conducive to cell adhesion. Therefore, we measured the change of contact angle of FP 6-4 before and after cleaning. The contact angle of the material before cleaning is 74.39 ± 2.84°, and the contact angle after cleaning is reduced to 24.48 ± 7.94° (Figure 6). The material contact angle before cleaning was slightly larger, which may be related to the high content of PCL and PEO. After cleaning, the pores between the fibers become larger, and the removal of PEO leads to the material’s hydrophilicity greatly improving.

As a biomaterial, it needs to have good mechanical properties, and the implanted material needs to be close to the physical and chemical properties of the repaired tissue [24], especially as a soft tissue repair material. According to the results in Table 3, it can be seen that the elastic modulus and tensile strength of the material decreased before and after cleaning, but the elongation of the material increased slightly. It may be related to the removal of PEO in the material and the dissolution of part of collagen.

The good pore structure of the material facilitates the transportation of nutrients and the adhesion and crawling of cells. It can be seen from Table 2 that the porosity of the material is significantly increased after cleaning. PEO is used as a sacrificial fiber to increase the porosity of the material. The material has preliminary cell proliferation, and crawling space structure.

### 3.3. Cell Culture

#### 3.3.1. Cell Adhesion

After 1d of co-culture, the cells could well aggregate and adhere to the surface of the material, and a pseudopod was formed. Cells can divide and proliferate on the surface of the material, and the pores between the fibers allow the pseudopod to extend into the material, making the bond between the cells and the material closer (Figure 7). These results showed that the cells grew well on the prepared FC-PCL nanofiber membrane material, which provided a good adhesion place for cells and was conducive to cell crawling and proliferation.

#### 3.3.2. Dead/Live Cell Text

The results of AMPI staining (Figure 8) showed that there were almost no dead cells in FP 6-4 and control group, indicating that the material had no obvious cytotoxicity. At 2d and 3d, there was no difference in the number of cells between the FP 6-4 material leaching solution group and the control group. At 5d, there was a great difference in the number of cells between the two groups. However, the cell density of FP 6-4 group was still within the normal range. The staining results showed that the cells had good cell morphology. The results can only preliminarily judge whether the material extract will affect cell proliferation, but its advantages and disadvantages still need to be further studied.

### 3.4. Safety and Degradation Tests In Vivo

In the results of HE staining (Figure 9), the undyed or lighter colored area is the material, and the darker colored area around the material is the inflammatory reaction zone. After 3d of implantation, the material was relatively complete, and the surrounding inflammatory cells were mainly multinucleated giant cells. After 7d of implantation, the cells in the material increased significantly. This result was also close to the experimental results of Hodge, J. et al. [25], and the cell infiltration increased significantly from the 7d. However, there were some necrotic cells around the material, which may be related to the engulfment of the material by the cells and the metabolism of the wound.

Relatively complete materials were still observed on 14d and 21d. Angiogenesis began in the material, neutrophils decreased, and lymphocytes increased in the early stage of tissue inflammatory reaction, and epithelioid cells appeared and gradually gathered to form multinucleated giant cells.

At 28d, some materials can still be seen, but the boundary between materials and tissues was no longer clear, and a large number of cells entered the materials. The implant material has no other abnormalities except normal inflammatory reaction, and can support cell infiltration and tissue vascularization. The results showed that the material has the potential as a biomedical material.

## 4. Conclusions

In this study, fish collagen was blended with PCL by electrospinning, and the nanofiber membrane material based on fish collagen was successfully prepared. The spinnability of the solution and the porosity of the material were improved by using the spinning aid PEO. The characterization and biological test of the material showed that the material had good mechanical properties and biocompatibility, can promote cell adhesion and proliferation, and can be used as a potential medical material for tissue defect and wound repair.

In addition, although people are currently interested in methods to increase the porosity of electrospinning scaffolds, these methods are rarely applicable to multi-component materials containing natural biological macromolecules such as collagen. The results of this study showed that electrospinning PEO sacrificial fibers was a relatively simple method, which can be used for electrospinning of collagen materials to increase the porosity of the material and cell infiltration.

In the biological test, this experiment only selected a representative material for the basic test of biological function, which proved that the preparation of fish collagen-based biomaterials by this method is safe and feasible, and can be used as a soft tissue repair material for further efficacy research. However, in the process of wound repair, due to the different recovery time of different wounds, the degradation time of the required materials is also different. This problem can be solved by adjusting the ratio of FC and PCL. Therefore, the next step is to choose an appropriate ratio to prepare FC-PCL nanofiber membrane for wound repair, and to determine its role and mechanism of wound repair. This research provides new ideas for the development and application of fish collagen.

## Figures and Tables

**Figure 1 nanomaterials-12-00900-f001:**
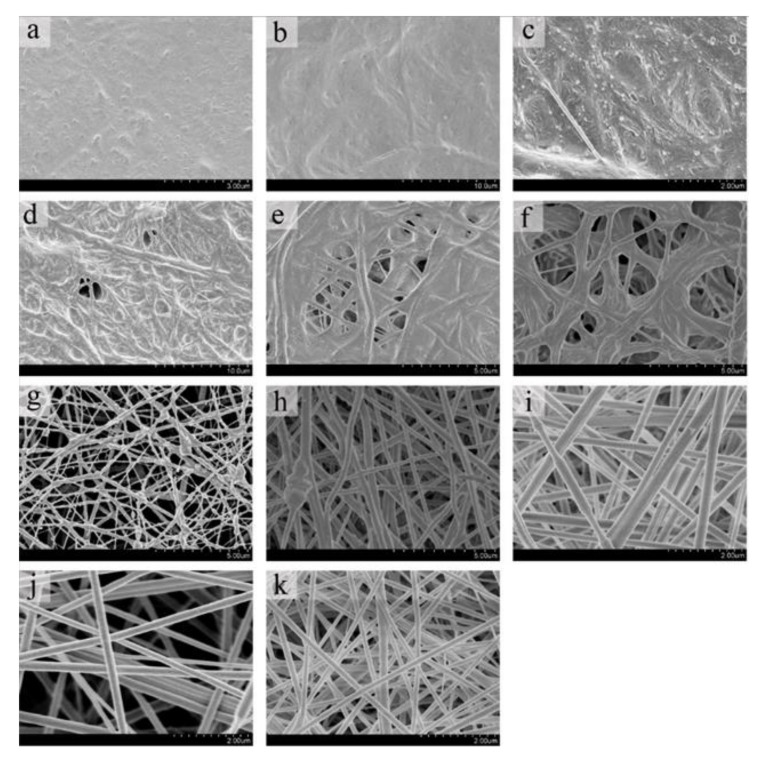
SEM results of FC-PCL nanofiber membrane. Figures (**a**–**k**) is the electrospinning results with m (FC): m (PCL) = 0: 10, 1:9, 2:8, 3:7, 4:6, 5:5, 6:4, 7:3, 8:2, 9:1, 10:0.

**Figure 2 nanomaterials-12-00900-f002:**
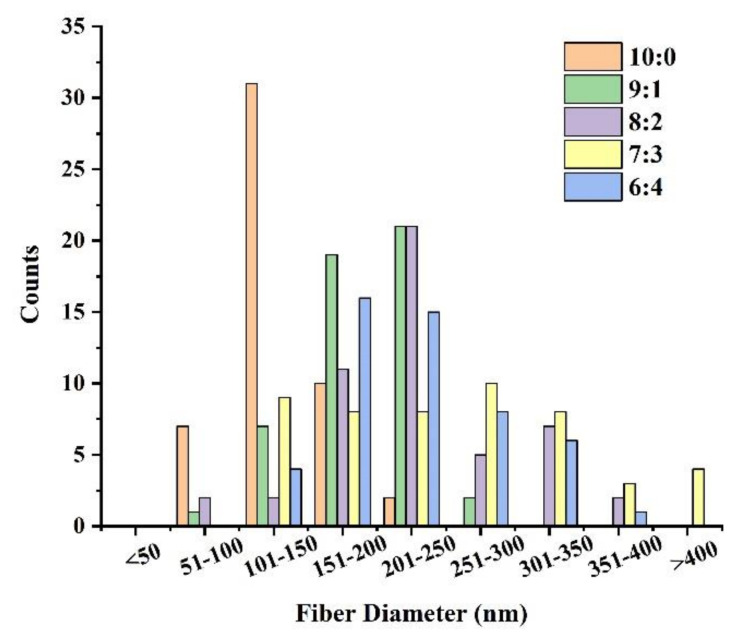
FC-PCL nanofiber membrane diameter distribution.

**Figure 3 nanomaterials-12-00900-f003:**
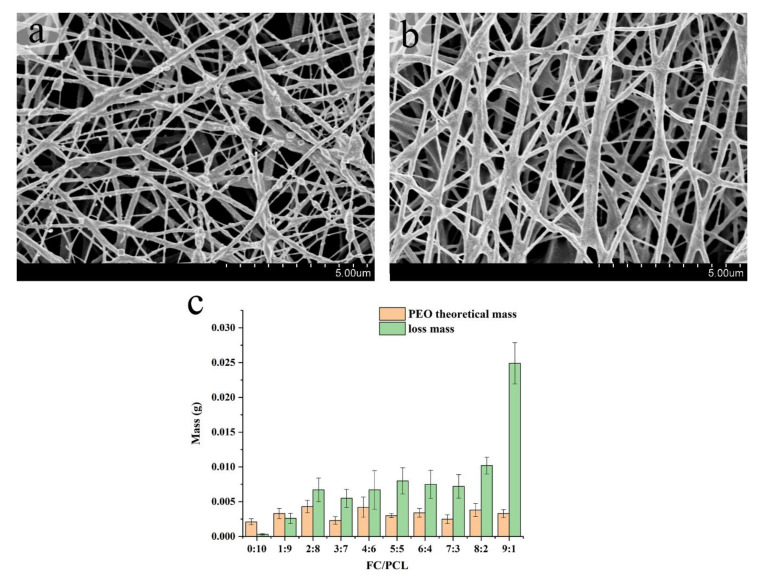
Results of PEO removal. (**a**) SEM image before PEO removal, (**b**) SEM image after PEO removal, (**c**) Mass loss of PEO removal.

**Figure 4 nanomaterials-12-00900-f004:**
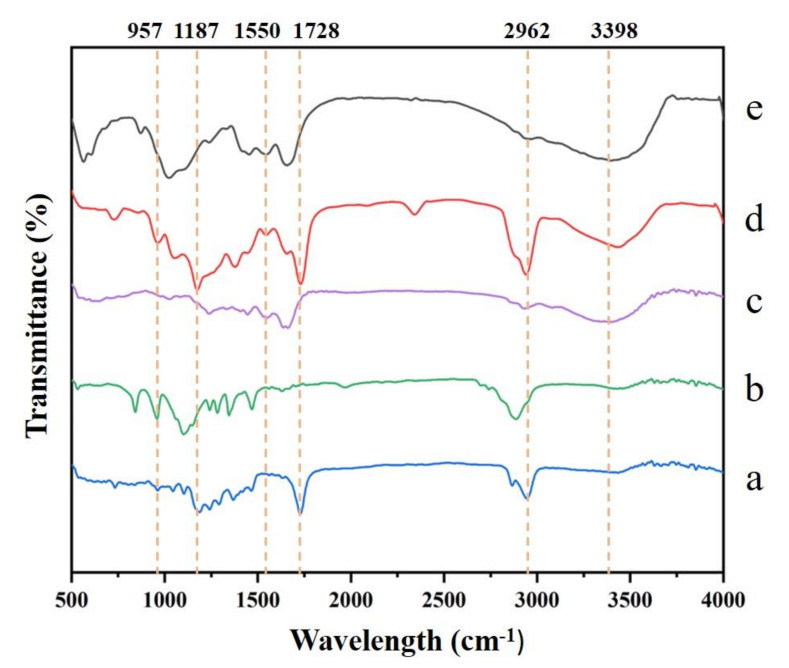
FTIR spectra of FC-PCL nanofiber membranes. (**a**) PCL, (**b**) PEO, (**c**) FC, (**d**) Before PEO removal, (**e**) After PEO removal.

**Figure 5 nanomaterials-12-00900-f005:**
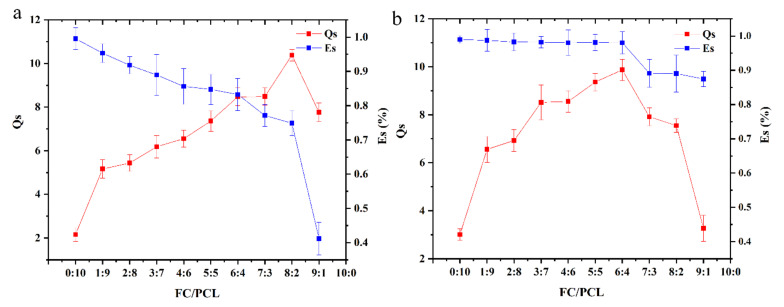
Swelling rate and hydrolysis stability of FC-PCL nanofiber membrane. (**a**) the swelling rate and hydrolysis stability before PEO removal (**b**) the swelling rate and hydrolysis stability after PEO removal.

**Figure 6 nanomaterials-12-00900-f006:**
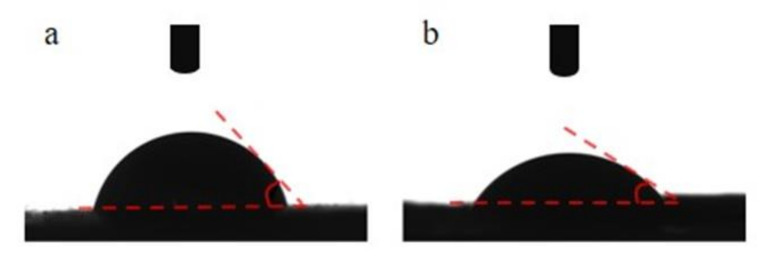
Contact angle. (**a**) Before PEO removal, (**b**) After PEO removal.

**Figure 7 nanomaterials-12-00900-f007:**
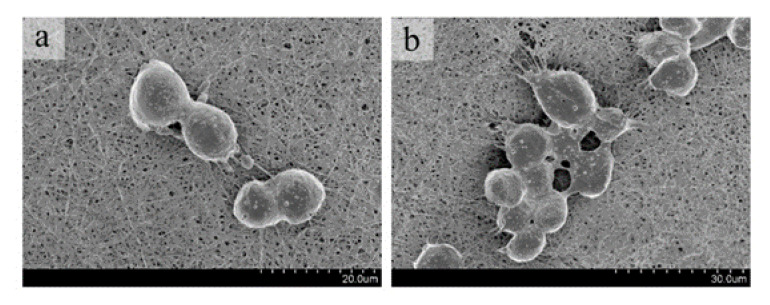
Cell-material co-culture. (**a**) cell division and proliferation, (**b**) cell adhesion.

**Figure 8 nanomaterials-12-00900-f008:**
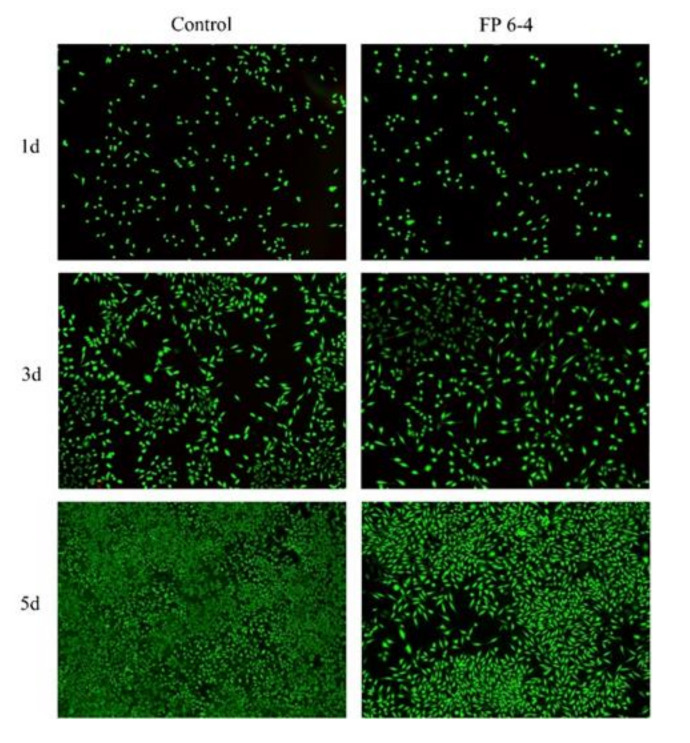
AMPI staining (green for living cells, red for dead cells).

**Figure 9 nanomaterials-12-00900-f009:**
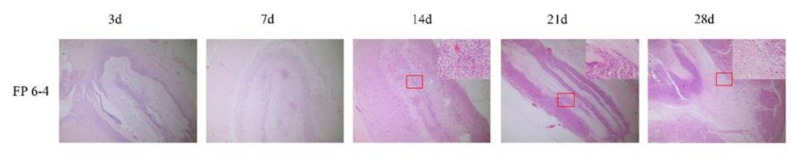
HE staining of degradation of FP 6-4 nanofiber membrane (scale bar: 1:40).

**Table 1 nanomaterials-12-00900-t001:** Effects of PEO content on spinning.

PEO Content (%)	Spinning Result
0.2	Unstable
0.4	Unstable
0.6	Film
0.8	Film
1.0	Unstable
1.2	Unstable, the needle was easy to block

**Table 2 nanomaterials-12-00900-t002:** Tentative assignments of FTIR Peaks for type I collagen, PCL, and PEO.

Polymer	Wavenumber (cm^−1^)	Assignment
FC	3398	amide A band, N-H stretching vibration
	1662	amide I band, C=O tensile vibration
	1550	amide II band, N-H and C-H bending
	1238	amide III band, C-O stretching vibration
PCL	2962 and 2859	CH_2_ stretching (asymmetric and symmetric)
	1728	C=O stretching
	1290	C-C stretching
	1241	C-O-C asymmetric stretching
PEO	2887	CH_2_ stretching
	1454	C-H bending
	1352, 1280 and 820	C-H deformation of the methyl group
	1174, 1053 and 957	C-O-C stretching vibration

**Table 3 nanomaterials-12-00900-t003:** Characterization of FP 6-4.

	Before PEO Removal	After PEO Removal
Contact angle (°)	74.39 ± 2.84	24.48 ± 7.94
Elongation (%)	10.67 ± 3.51	13.67 ± 1.15
Tensile strength (MPa)	145.57 ± 44.08	112.45 ± 22.34
Elastic Modulus (MPa)	4635.33 ± 579.50	2751.00 ± 426.18
Porosity (%)	61.56 ± 3.916	87.6 ± 9.273

## Data Availability

The data presented in this study are available in this article.

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
