# Peer review of "Polyethylene Oxide Assisted Fish Collagen-Poly-ε-Caprolactone Nanofiber Membranes by Electrospinning"

_nanomaterials, 2022, doi:10.3390/nano12060900_

Round 1

Reviewer 1 Report

The interesting point of this study is as follows: 

Although many studies are currently interested in methods to increase the porosity of electrospinning scaffolds, these methods are rarely applicable to multi-component materials containing natural biological macromolecules such as collagen. The results of this study show that electrospinning PEO sacrificial fibers is a relatively simple method, which can be used for electrospinning of a variety of materials containing collagen to increase cell infiltration. 

The paper proved that the preparation of fish collagen-based biomaterials by this method is safe and feasible, and can be used as a soft tissue repair material for further efficacy research. However, it is not sufficient data to ensure the biological study including functional characterization of cell culture such as gene and protein analysis, and  in vivo study.

Author Response

Dear Editors and Reviewers:

Thank you for your letter and for the reviewers' comments concerning our manuscript entitled“PEO assisted fish collagen-PCL nanofiber membranes by electrospinning" (nanomaterials-1563880). Those comments are all valuable and very helpful for revising and improving our paper, as well as the important guiding significance to our researches. We have studied comments carefully and have made correction which we hope meet with approval.

Revised portion are marked in yellow in the paper. The main corrections in the paper and the responds to the reviewer' s comments are as flowing:

Responds to the reviewer 's comments:

Reviewer #1:

  1. Response to comment: The materials mentioned in this paper have the potential of biomedical materials, which is based on the analysis of the experimental results of cytotoxicity and degradation in vivo. Firstly, we tested the cytotoxicity of the material through AMPI experiment and proved that the material has no cytotoxicity. Secondly, we co cultured the cells with the material to prove that the material supports cell adhesion and proliferation. Finally, in vivo degradation test showed that the degradation rate of the material in vivo was about one month, and the material could support cell infiltration and vascularization. These results preliminarily prove that the materials we prepared have the potential as biomaterials. As for the functional characterization of cell culture and in vivo research, this is also our next research direction. These experiments are ongoing and need to be verified by repeated experiments. Therefore, the function of the material may not be fully demonstrated in this manuscript.

Thank you again for your valuable comments and attention to our manuscript.

Best Regards.
Yours sincerely,

Ms. Xiaoli He

Reviewer 2 Report

The manuscript describes the design of a electrospun nanofibrous scaffolding made of a combination of fish-collagen and PCL to study cell crawling and proliferation, as inflammatory process and eventual toxicity. To optimize some electrospinning process features, the polymer solution was added with PEO, as sacrificial material. Unfortunately the text is confused and lacks numerous data and information, therefore, in conclusion, neither the objectives nor the results look very clear. Below is a list of some comments, questions and suggestions to the Authors:

  1. 28-30: the statements here reported and related to the advantages in using fish-collagen instead of collagen from other sources, must be confirmed by bibliographic references.
  2. 60: would Authors explain why collagen has good mechanical energy?
  • 141: instrument and software to calculate WCA should be mentioned?
  1. 181: Authors previously declared that the best FC:PCL ratio was 8:2 in order to test the appropriate PEO amount between 0.2% and 0.12%, but the selected percentage of PEO was 8% (?).
  2. In Figure 1 several SEM images are reported as fish-collagen increases compared to PCL. Since PEO is presumably inside, it should be mentioned. Furthermore, the measuring bars are illegible.
  3. Could Authors provide any SEM micrograph concerning FC-PCL completely without PEO in order to stress the contribution of PEO to design nanofibers? Similarly, what about fibers morphologies when PEO is dissolved?
  • 229 and R.238: correct punctuation
  • Figure 3 must be explained. There are not sufficient information to understand its meaning (PEO theritical quality?) and what data they used to calculate the theoretical quality of the PEO. Any pictures about that, in order to aid their statements?
  1. From Table 2 all the measured parameters decreased. Could the Authors explain why they stated (R.334) that elongation of the material increased slightly despite the other features?
  2. 321 and R.327: Angle with lowercase
  3. Figure 6: before PEO cleaning the contact angle doesn’t look strongly hydrophobic as the Authors stated. From Figure, I’m not sure the measured angle is correctly outlined. It should be interesting if Authors compared WCA of similar nanofibers also without PEO treatment in order to enhance the PEO effects (which is more hydrophilic than PCL…up to be solved).
  • Figure 7: Could Authors spend more words about the selected material for cell-culture?
  • Figure 8: what the Authors mean for control? Could they explain what is AMPI? Why they did describe only fiber FP6-4? What happened to the others?
  • HE staining should be referenced and shortly described

Author Response

Dear Editors and Reviewers:

Thank you for your letter and for the reviewers' comments concerning our manuscript entitled“PEO assisted fish collagen-PCL nanofiber membranes by electrospinning" (nanomaterials-1563880). Those comments are all valuable and very helpful for revising and improving our paper, as well as the important guiding significance to our researches. We have studied comments carefully and have made correction which we hope meet with approval.

The confused part of the manuscript has been revised, and revised portion are marked in yellow in the paper. The main corrections in the paper and the responds to the reviewer' s comments are as flowing:

Responds to the reviewer 's comments:

Reviewer #2:

1.28-30: The manuscript is supplemented with literature confirming the advantages of using fish collagen instead of other sources of collagen.

2.58-60: The collagen-PCL composite membrane prepared has both the excellent biological properties of collagen and the good mechanical energy of PCL. Collagen has poor mechanical properties, so it needs to be compounded with PCL to prepare composites with good mechanical properties.

  1. 87, 143: The instrument and software for measuring WCA are supplemented. The contact angles of different spinning films were detected by SDC-500 contact angle measuring instrument and water drop shape analysis system.

4.206-214: I'm sorry that I made a mistake in the result analysis due to my negligence. Now I have corrected the content in the manuscript.

5-6. As for the modification and supplement of SEM images, our SEM needs to be carried out in the public test center. But now it's their break during the spring festival break and it's not open to the public. The resumption time is February 15, 2022, so I may need some time to complete the modification of SEM.

7.231, 240: Punctuation has been modified.

8.117-118, 243-244: The calculation formula of PEO theoretical mass and sample loss mass has been added, and figure 3 has also been modified.

  1. According to the results in Table 2, it can be seen that the elongation of the material increased from 10.67 ± 3.51 to 13.67 ± 1.15.
  2. Confirmed

11.336: The expression of strong hydrophobicity here is really not rigorous enough. We want to say that the hydrophilicity of the material is improved through the removal of PEO. Therefore, the content of the manuscript has been modified: The material contact angle before cleaning was slightly larger, which may be related to the high content of PCL and PEO.

  1. Figure 7:The selected co culture materials were supplemented. 318-322

Biomaterials should have good biological properties, mechanical properties and spatial structure to support cell growth. In addition, the implanted material should also have certain hydrolysis stability to prevent excessive degradation in vivo. Considering the above factors, we selected M (FC): m (PCL) = 6:4 (FP 6-4) nanofiber membrane as the representative material for biological evaluation and characterized it.

  1. The control group is a group of cells cultured in DMEM medium, which forms a control with the experimental group cultured in material extract to detect the cytotoxicity of materials. In previous optimization experiments, it was found that many materials with FC and PCL ratios are not suitable for biomaterials. Therefore, the optimized material fp6-4 is selected as the representative of the subsequent test, and the results will be relatively concise and clear. Considering the consistency of the results before and after, the manuscript only shows the cytotoxicity test results of the materials, which proves that the materials prepared by this method have no cytotoxicity.

14.186-187: The contents and references of HE staining have been supplemented.

Thank you again for your valuable comments and attention to our manuscript.

Best Regards.
Yours sincerely,

Ms. Xiaoli He

Reviewer 3 Report

The authors have reported bio-compatible nanofiber preparation methods in the manuscript. They facilitated PEO polymer as a sacrificial layer to improve mechanical and adhesion property of the collagen base material for electrospinning. Since the reported work is containing useful information for the researchers who sought to prepare biocompatible nanofiber based membrane, I recommend the manuscript to be published in 'nanomaterials' after some of major revisions that are suggested in the below.

1) Use of abbreviations, such as PEO and PCL, usually are not recommeded in title.
2) In line 68, 69, I presume that 'PEO fibers' are typo which are intended to be 'PCL fiber'
3) Since one of the main issues of the manuscript is the role of PEO for nanofiber preparation, it would be informative if the authors can show what is happening when PEO is not included in FC-PCL nanofiber prepartion process.

4) Regarding FTIR spectroscopy on their prepared sample (Figure 4), too many peak assignements, which may be non-relevant to prove PCL is removed by washing, can be confusing to possible readers.
I recommed the authors to include table for peak assignements to improve the readability. Also, relevant papers should be referred for every spectra peaks, if authors intend to use the specific peaks for their arguments. 

5) Since the peak at ~2887 cm-1, which the author assigned to CH2 stretching vibration, exist in both PEO and PCL materials (Figure 4a, b), I do not understand why the disappearance of the peak can support the author's argument that PEO was removed away thoroughly after washing. Peak-to-peak correlation study or peak width modification due to the process may be helpful to clarify the argument.

In brief, the manuscript need to address the above-mentioned concerns to be published in 'nanomaterials'.

Author Response

Dear Editors and Reviewers:

Thank you for your letter and for the reviewers' comments concerning our manuscript entitled“PEO assisted fish collagen-PCL nanofiber membranes by electrospinning" (nanomaterials-1563880). Those comments are all valuable and very helpful for revising and improving our paper, as well as the important guiding significance to our researches. We have studied comments carefully and have made correction which we hope meet with approval.

Revised portion are marked in yellow in the paper. The main corrections in the paper and the responds to the reviewer' s comments are as flowing:

Responds to the reviewer 's comments:

Reviewer #3:

  1. According to your suggestion, change the abbreviations of PEO and PCL in the title to the polyethylene oxide and poly-ε-caprolactone. The revised title is: Polyethylene oxide assisted fish collagen-poly-ε-caprolactone nanofiber membranes by electrospinning.
  2. 68-89: This section described the solubility of PEO fibers, so it can be removed as sacrificial fibers by cleaning.
  3. PEO plays two main roles in this experiment. One is to improve the spinnability of the solution. When PEO is not added, many solutions with FC and PCL ratio cannot obtain membrane materials. The second function is to improve the porosity of the material as a sacrificial template, which is shown in Table 2. A description of the role of PEO is also added to the result analysis of this manuscript.195-205
  4. References to spectral peaks have been added
  5. After material cleaning, the absorption peak of CH2 stretching vibration of PEO be-comes smaller, and the absorption peak of C-O-C stretching vibration disappears, indicating that the removal of PEO was relatively thorough. The change of these peaks, combined with the mass loss before and after cleaning, speculates that the removal of PEO is relatively complete, but this result is not absolute.

Thank you again for your valuable comments and attention to our manuscript.

Best Regards.
Yours sincerely,

Ms. Xiaoli He

Round 2

Reviewer 2 Report

The manuscript looks improved after Authors' revision, but their replies  sometimes seem very approximate. First of all, I would like to give the Authors a suggestion: it would have been proper to rewrite the questions and reply to each one immediately afterwards instead of just producing a list of answers leaving the reviewer to retrieve his/her list of questions. 
Returning to the quality of their replies, they were often not comprehensive and complete.
1) SEM images in a scientific work require the measurement bar to allow visual comparison. In addition, the required control sample and the morphology of the fibers would be missing at the dissolution of the PEO. This is a scientifically useful information for understanding the work results. The explanation that the SEM office is closed is not acceptable! However, now, it should be open (according to the info provided by Authors).
2) Authors should add a literary reference to the term theoretical quality of the PEO, because unfortunately the meaning of that and its related calculation (multiplication) in the definition continues to be dark to me.
3) "Angle" term has to be written with the lowercase letter… it has not been replicated nor modified
4) WCA images with PCO-free fibers have not been added and the description highlighted in yellow referring to the scientific literature is still confusing (should be revised)
5) Regarding the cells used, why do they think FP6-4 are the most representative? Authors should be more detailed in their replies.

Author Response

Dear Editors and Reviewers:

Thank you again for your letter and for the reviewers' comments concerning our manuscript entitled“PEO assisted fish collagen-PCL nanofiber membranes by electrospinning" (nanomaterials-1563880). Those comments are all valuable and very helpful for revising and improving our paper, as well as the important guiding significance to our researches. We have studied comments carefully and have made correction which we hope meet with approval.

The problems of the manuscript were revised in the review mode. The main corrections in the paper and the responds to the reviewer' s comments are as flowing:

Responds to the reviewer 's comments:

Reviewer #2:

  1. SEM images in a scientific work require the measurement bar to allow visual comparison. In addition, the required control sample and the morphology of the fibers would be missing at the dissolution of the PEO. This is a scientifically useful information for understanding the work results. The explanation that the SEM office is closed is not acceptable! However, now, it should be open (according to the info provided by Authors).

Respond:All SEM images are exported from the instrument with a default scale, which I think is very accurate, so no measurement bar is added. If you think this is not clear enough, I am willing to modify all SEM images again. As for the SEM images before and after PEO removal, I'm sorry. The deadline for the last modification is the last day of the holiday of our public test center. Therefore, there was no time to make changes in the final reply. This time, the comparative pictures (Fig. 3, a and b) and the description of the pictures were added.

  1. Authors should add a literary reference to the term theoretical quality of the PEO, because unfortunately the meaning of that and its related calculation (multiplication) in the definition continues to be dark to me.

Respond:The theoretical mass calculation formula of PEO was modified

  1. "Angle" term has to be written with the lowercase letter… it has not been replicated nor modified

Respond:All contact “Aangle” were changed to lowercase contact angle

  1. WCA images with PCO-free fibers have not been added and the description highlighted in yellow referring to the scientific literature is still confusing (should be revised)

Respond:The material we prepared needs PEO spinning aid to obtain a complete membrane. I don't know whether the contact angle of the material without PEO refers to the removal of PEO after preparation (Fig.6, b), or PEO was not added during the preparation (the solution is unstable).

  1. Regarding the cells used, why do they think FP6-4 are the most representative? Authors should be more detailed in their replies.

Respond:251-254 Added description of the selection of the FP6-4 material.

Thank you again for your valuable comments and attention to our manuscript.

Best Regards.
Yours sincerely,

Ms. Xiaoli He

Reviewer 3 Report

The authors have addressed most of reviewer's comment, however, interpretation on Figure 4 is not clear, yet. 

The purpose of FTIR spectra measurement on the film seems to prove that cleaning process has been thorough to remove PEO. Disappear of peak around 2887 cm-1 by cleaning, which is assigned to CH, is easy to find, but peaks related to C-O-C stretching is hard to interpret mainly due to overlap of many vibration modes at that specific spectral window. 

1) Can you perform more rigorous quantitative analysis on the peak. For example, peak intensity analysis after background subtraction would give more clear interpretation.

2) Can you point the important spectral peaks (C-O-C, CH2) in Figure 4, to improve the readability of the manuscript?

Author Response

Dear Editors and Reviewers:

Thank you again for your letter and for the reviewers' comments concerning our manuscript entitled“PEO assisted fish collagen-PCL nanofiber membranes by electrospinning" (nanomaterials-1563880). Those comments are all valuable and very helpful for revising and improving our paper, as well as the important guiding significance to our researches. We have studied comments carefully and have made correction which we hope meet with approval.

The problems of the manuscript were revised in the review mode. The main corrections in the paper and the responds to the reviewer' s comments are as flowing:

Responds to the reviewer 's comments:

Reviewer #3:

  1. Can you perform more rigorous quantitative analysis on the peak. For example, peak intensity analysis after background subtraction would give more clear interpretation.

Respond: The results have been reanalyzed according to the change of relative intensity of absorption peak.

  1. Can you point the important spectral peaks (C-O-C, CH2) in Figure 4, to improve the readability of the manuscript?

Respond: In order to improve the readability of the manuscript, Fig.4 was modified and important spectral peaks were listed (Table 2)

Table 2. Tentative assignments of FTIR Peaks for FC, PCL, and PEO (added)

Polymer

Wavenumber (cm–1)

Assignment

FC

3398

amide A band, N-H stretching vibration

1662

amide I band, C=O tensile vibration

1550

amide II band, N-H and C-H bending

1238

amide III band, C-O stretching vibration

PCL

2962 and 2859

CH2 stretching (asymmetric and symmetric)

1728

C=O stretching

1290

C-C stretching

1241

C-O-C asymmetric stretching

PEO

2887

CH2 stretching

1454

C-H bending

1352, 1280 and 820

C-H deformation of the methyl group

1174, 1053 and 957

C-O-C stretching vibration

Thank you again for your valuable comments and attention to our paper.

Best Regards.
Yours sincerely,

Ms. Xiaoli He

Round 3

Reviewer 2 Report

The manuscript has been improved and the Authors replied properly to the questions accordingly. My suggestion is to consider their work suitable for publication.

Kind regards

Antonella

Reviewer 3 Report

The manuscript has been improved significantly after the revision. I would recommend the manuscript to be published in 'Nanomaterials' as it is.